# Synthesis, structures and magnetic properties of [($\eta^9$-C$_9$H$_9$)Ln($\eta^8$-C$_8$H$_8$)] super sandwich complexes

L. Münzfeld [1], C. Schoo [1], S. Bestgen [1], E. Moreno-Pineda [2], R. Köppe [1], M. Ruben [2,3] & P.W. Roesky [1]

Sandwich complexes are an indispensable part of organometallic chemistry, which is becoming increasingly important in the field of lanthanide-based single molecule magnets. Herein, a fundamental class of pure sandwich complexes, [($\eta^9$-C$_9$H$_9$)Ln($\eta^8$-C$_8$H$_8$)] (Ln=Nd, Sm, Dy, Er), is reported. These neutral and sandwiched lanthanide compounds exclusively contain fully $\pi$-coordinated coplanar eight and nine membered CH rings. The magnetic properties of these compounds are investigated, leading to the observation of slow relaxation of the magnetization, including open hysteresis loops up to 10 K for the Er(III) analogue. Fast relaxation of the magnetization is likewise observed near zero field, a highly important characteristic for quantum information processing schemes. Our synthetic strategy is straightforward and utilizes the reaction of [($\eta^8$-C$_8$H$_8$)LnI(thf)$_n$] complexes with [K(C$_9$H$_9$)]. Although all compounds are fully characterized, structural details of the title compounds can also be deduced by Raman spectroscopy only.

[1] Institute of Inorganic Chemistry, Karlsruhe Institute of Technology (KIT), Engesserstraße 15, 76131 Karlsruhe, Germany. [2] Institute of Nanotechnology (INT), Karlsruhe Institute of Technology (KIT), Hermann-von-Helmholtz-Platz 1, D-76344 Eggenstein-Leopoldshafen, Germany. [3] Institut de Physique et Chimie des Matériaux de Strasbourg (IPCMS), CNRS-Université de Strasbourg, 23 rue du Loess, BP 43, F-67034, Strasbourg Cedex 2, France. Correspondence and requests for materials should be addressed to E.M.-P. (email: eufemio.pineda@kit.edu) or to M.R. (email: mario.ruben@kit.edu) or to P.W.R. (email: roesky@kit.edu)

Sandwich complexes, that is, compounds bearing exclusively two planar, cyclic and π-bonded ligands, are a fundamental class of compounds in organometallic chemistry. In fact, the discovery of ferrocene $[(\eta^5-C_5H_5)_2Fe]$ by Kealy and Pauson[1] and the subsequent structural analysis by Fischer and Wilkinson paved the way to modern organometallic chemistry[2,3]. Ever since, the quest for new sandwich complexes has been a central part of modern organometallic chemistry. In the last decades, sandwich complexes, and particularly ferrocene, have become widely used compounds, which found a variety of applications, e.g. in synthesis, catalysis, electrochemistry, medicine and even as fuel additive[4]. In a classical homoleptic sandwich or metallocene complex, two identical aromatic ring systems equally bind with all carbon atoms to a metal center[5]. Well established examples of this structural motif are the above-mentioned ferrocene, bis(benzene) chromium and uranocene[1,6,7]. Besides these homoleptic complexes, there are also examples of heteroleptic sandwich complexes ligated by two different aromatic ring systems. As aromatic moieties, rings ranging from three to nine-membered systems have been established in organometallic chemistry. Despite the large variety of possible ligand permutations, considering these seven different ring sizes, only a limited number of structurally characterized ligand combinations has been reported in terms of homoleptic and heteroleptic complexes. These are the four non-substituted homoleptic metallocene archetypes with five to nine-membered rings: $[(\eta^5-C_5H_5)_2M]$[1–3], $[(\eta^6-C_6H_6)_2M]$[7], $[(\eta^8-C_8H_8)_2M]$[6,8], and $[(\eta^9-C_9H_9)_2M]$ (Fig. 1)[9–11]. Considering heteroleptic sandwich complexes, only four non-substituted types were structurally characterized, which are: $[(\eta^5-C_5H_5)M(\eta^4-C_4H_4)]$[12], $[(\eta^6-C_6H_6)M(\eta^5-C_5H_5)]$[13,14], $[(\eta^7-C_7H_7)M(\eta^5-C_5H_5)]$[15–18], and $[(\eta^8-C_8H_8)M(\eta^5-C_5H_5)]$ (Fig. 1)[18–20]. All of these are cyclopentadienyl derivatives combined with four to eight-membered rings.

Obviously, the vast majority of sandwich complexes is ligated by cyclopentadienyl derived moieties. In contrast, complexes ligated by larger aromatic monocycles are scarce. Therefore, we define the class of sandwich complexes having more than 16 carbon atoms coordinated to the central metal atom as super sandwich compounds, to distinguish them from classical sandwich complexes.

One of the most recent application of sandwich compounds in rare earth chemistry is their use as single molecule magnets (SMMs)[21–30]. Such lanthanide-based SMMs have been shown to act as quantum computing units, so-called qudits, for the implementation of Grover´s quantum search algorithm[31], and more recently have displayed magnetic hysteresis at liquid nitrogen temperatures[32–34].

A popular model for the SMM behavior of mononuclear lanthanide complexes focuses on the stabilization of the corresponding lanthanide ions $m_J$ ground state by tuning the local electron density around the lanthanide ion generated by the ligand sphere[27]. Two prominent examples proving this concept are dysprosium and erbium. For example, the highest $m_J$ state (±15/2) for dysprosium(III) has an oblate shape, thus an axial ligand field enhances the anisotropic properties of dysprosium containing complexes[35]. Recently, significant advances were reported by using homo- and heteroleptic cyclopentadienyl based dysprosium(III) metallocene cations, which exhibit a highly axial ligand field, enabling record high anisotropy barriers[32–34]. On the other hand the highest $m_J$ state of erbium(III) is prolate shaped, therefore, an equatorial ligand field is beneficial in this case. This can be achieved by introducing one or two $\eta^8-C_8H_8$ ligands, which exert a strong equatorial ligand field, into the coordination sphere of erbium ions[27,28,35–37]. These two examples highlight that the local symmetry generated around the central lanthanide ion, determined by the ligand field and the rigidity of the complex, plays a crucial role in the design of SMMs[23,24,38–42]. A review published recently pointed out that other, uncommon ligand systems, such as the cyclononatetraenyl anion may shed light on interesting properties in terms of SMM behavior and fundamental magneto-structural correlations[23].

Herein, we present a long sought for class of sandwich complexes $[(\eta^9-C_9H_9)Ln(\eta^8-C_8H_8)]$, which exclusively contain fully π-coordinated eight and nine-membered rings. Synthesizing these compounds was already attempted by Streitwieser et al.[43] in 1973, shortly after the first successful synthesis of $KC_9H_9$ was reported by Katz and coworkers[43,44]. Their strategy was based on a one-pot reaction between $LnCl_3$ (Ln=Ce(III), Pr(III), Nd(III), Sm (III)), $K_2C_8H_8$, and $KC_9H_9$. However, they could only isolate complexes of the type $[(\eta^8-C_8H_8)LnCl(thf)_2]$ thereby highlighting, that the $C_9H_9^-$ anion does not form sandwich complexes analogous to $C_8H_8^{2-}$. After a 45 years quest for $[(\eta^9-C_9H_9)Ln(\eta^8-C_8H_8)]$, we now report a synthetic protocol based on two distinct steps.

## Results

**Synthesis and crystallographic characterization of $[(\eta^9-C_9H_9)Ln(\eta^8-C_8H_8)]$.** First, we synthesized the starting material $KC_9H_9$ following the procedure of Katz et al.[44] The $^1H$-NMR spectrum shows only one sharp resonance at δ 7.05 ppm, which is attributed to the nine ring protons and consistent with the regular **1**-all-cis-configuration being present in solution. Additionally, the molecular structure of the dimethoxyethane solvate $[(\eta^9-C_9H_9)K(DME)_2]$ (**1**) was established by X-ray diffraction experiments. A flat and aromatic nine-membered carbon ring is observed with C–C bond lengths ranging from 1.389(3) Å to 1.394(3) Å, which is in the expected region for aromatic $sp^2$-hybridized carbon atoms (Fig. 2). Only the perfectly nonagonal all cis-isomer was found in the solid state and no positional disorder, indicating the presence of the cis,cis,cis,trans-cyclononatetraenyl isomer, was observed. This is in contrast to very recent findings from Nocton et al.[11], who also reported on the solid-state structure of $[(\eta^9-C_9H_9)K(OEt_2)]$. They obtained $KC_9H_9$ from diethyl ether as a mixture of cis- and trans-isomers of the $C_9H_9^-$ ring and discussed the influence of the isomer on its subsequent reactivity. The potassium ion is centered below the ring and shows a complete $\eta^9$-coordination with K–C bond distances ranging from 3.085(2)–3.154(2) Å.

With $KC_9H_9$ in hand, we next aimed to synthesize defined $[(\eta^8-C_8H_8)LnI]$ complexes, in which the residual iodide ligand can be replaced by $C_9H_9^-$ in a salt metathesis approach. Therefore, we synthesized $[(\eta^8-C_8H_8)LnI(thf)_n]$ (Ln=Nd(III)

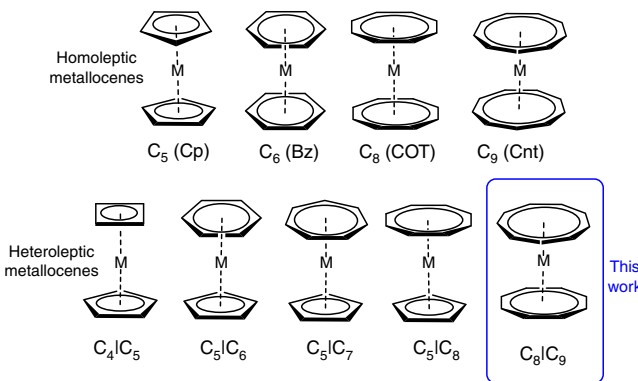

**Fig. 1** Examples of sandwich complexes. Structurally characterized homoleptic and heteroleptic metallocenes with $C_x$ ring systems (x = 4-9; M = metal atom)

(**2a**), Sm(III) (**2b**), Dy(III) (**2c**), Er(III) (**2d**); $n = 2$ (Sm, Dy, Er), 3 (Nd)) according to a facile synthetic protocol reported by Mashima et al.[45], which is based on the direct reaction of the lanthanide metal, cyclooctatetraene and iodine in hot THF (Fig. 3).

Compounds **2a** and **2b**, which have been reported earlier by Mashima et al.[45], were obtained within two days. Complex **2c** and **2d** could only be obtained on this route after activation of the

metal by in situ amalgamation. Nevertheless, significantly longer reactions times (3–4 weeks) were needed to obtain crystalline yields of 57% (**2c**) and 43% (**2d**). Ultimately, we were able to react the $[(\eta^8\text{-}C_8H_8)LnI(thf)_2]$ complexes with **1** in refluxing toluene, which gave the title compounds $[(\eta^9\text{-}C_9H_9)Ln(\eta^8\text{-}C_8H_8)]$ in moderate crystalline yields of 36% (Nd, **3a**), 32% (Sm, **3b**), 31% (Dy, **3c**) and 32% (Er, **3d**) (Fig. 4).

Single crystals of the heteroleptic sandwich complexes $[(\eta^8\text{-}C_8H_8)Ln(\eta^9\text{-}C_9H_9)]$ (Ln=Nd (**3a**), Sm (**3b**), Dy, (**3c**), Er (**3d**)) were obtained from toluene. The solid-state structures of **3c** and **3d** show a disorder of the eight and the nine-membered rings (see Supplementary Information for details). Especially the molecular solid-state structure of **3d** exhibits a complicated disorder with split positions of the Er(III) ion, showing slight indications of ring slipping and tilting in both ligands. The Er-C distances in the eight-membered ring vary between 2.406(11) Å and 2.670(10) Å with 5 carbon atoms closer located to the Er(III) ion than the others. Similarly, the Er-C distances in the nine-membered ring vary between 2.468(12) Å and 2.733(10) Å with four carbon atoms in closer proximity to the Er(III) ion (see Supplementary Table 10 for detailed bond lengths). However, this might also be caused by the unusual split Er(III) positions and thus, do not reflect the actual binding mode of the two aromatic moieties. We, therefore, performed a DFT geometry optimization and found the energetic minimum for **3d** to be a perfect sandwich-type configured $[(\eta^9\text{-}C_9H_9)Er(\eta^8\text{-}C_8H_8)]$ molecule (see Fig. 5). As a result, we propose **3d** to comprise two fully π-coordinated and

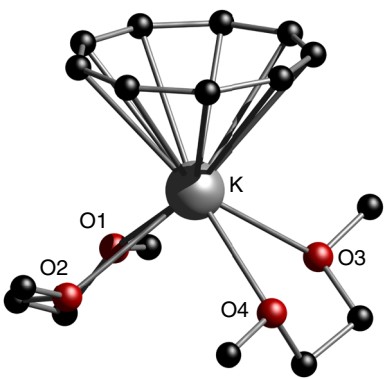

**Fig. 2** Molecular solid-state structure of compound **1**. $[(\eta^9\text{-}C_9H_9)K(DME)_2]$ **1**. Color code: K, gray; C, black; O, red. Hydrogen atoms are omitted for clarity

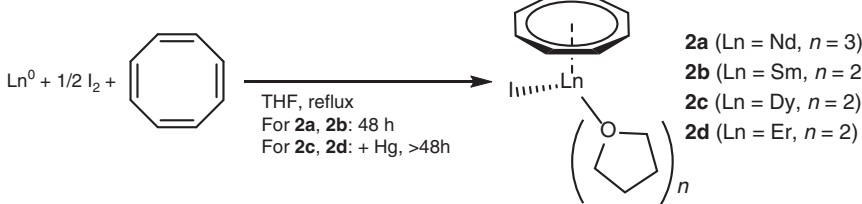

**Fig. 3** General synthetic procedure part 1. Synthesis of the cyclooctatetraene-complexes $[(\eta^8\text{-}C_8H_8)LnI(thf)_n]$ (Ln = Nd (**2a**), Sm (**2b**), Dy (**2c**), Er (**2d**))

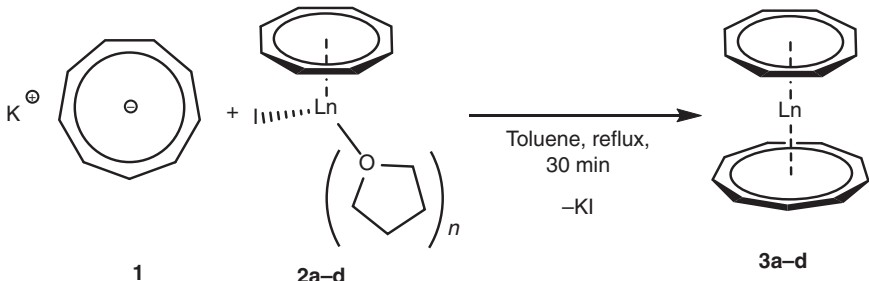

**Fig. 4** General synthetic procedure part 2. Synthesis of the heteroleptic sandwich complexes $[(\eta^9\text{-}C_9H_9)Ln(\eta^8\text{-}C_8H_8)]$ (Ln = Nd (**3a**), Sm (**3b**), Dy (**3c**), Er (**3d**); $n = 2$ (Sm, Dy, Er), 3 (Nd))



**Fig. 5** Molecular structures of compounds **3a**–**d**. $[(\eta^9\text{-}C_9H_9)Nd(\eta^8\text{-}C_8H_8)]$ **3a** (left) and $[(\eta^9\text{-}C_9H_9)Sm(\eta^8\text{-}C_8H_8)]$ **3b** (middle) in the solid state. Only on part of the disordered structures is depicted here. Geometry optimized structure of **3d** (right). Color code: H, light; C, black; Nd, cyan; Sm, pale green; Er, orange

**Table 1 Ln-Cg distances in the solid state of compounds 3a–d**

| Compound | Ln-Cg($C_8H_8$) [Å] | Ln-Cg($C_9H_9$) [Å] |
|---|---|---|
| 3a | 1.8925 (3) | 2.0441 (3) |
| 3b | 1.8687 (5) | 1.9908 (6) |
| 3c | 1.8869 (4) | 1.8752 (3) |
| 3d | 1.6725 (4) | 1.7248 (4) |

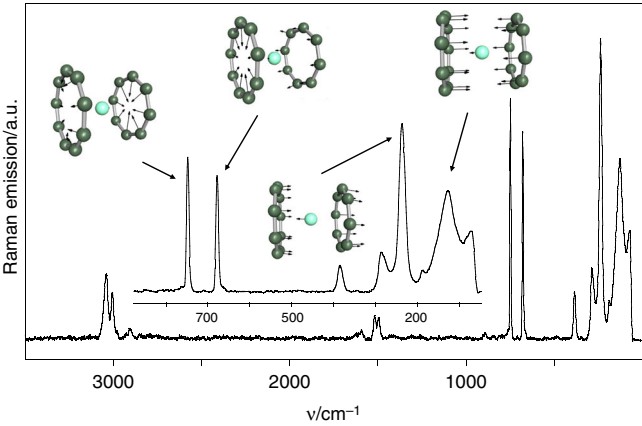

**Fig. 6** Raman-spectroscopic analysis. FT-Raman spectrum of [($\eta^9$-$C_9H_9$)Sm($\eta^8$-$C_8H_8$)] **3b**. The motion vectors of four prominent bands are inserted. Hydrogen atoms and their motion vectors are omitted for clarity

coplanar aromatic ligands, although the solid-state structure does not undoubtedly proof this assumption. On the other hand, **3a** and **3b** did not show this type of disorder. Their solid-structures exhibit perfect sandwich-type configuration with both rings bound to the central lanthanide ion in a coplanar fashion (Fig. 5).

Only the structural parameters of **3a** are discussed here in detail (Ln-Cg distances for **3a–d** are given in Table 1). In compound **3a**, the neodymium atom is centered between both rings with Nd-C bond lengths of 2.613(8)–2.653(8) Å to the eight-membered ring and Nd-C bond lengths of 2.845(6)–2.915 (7) Å to the nine-membered ring. The distances of the Nd atom to the ring centroids (Cg) are Nd-Cg(8) 1.8925(3) Å and Nd-Cg (9) 2.0441(3) Å. The Cg(8)-Nd-Cg(9) angle accounts for 176.47 (1)° and underpins an almost ideal coplanar arrangement. Interestingly, the $\eta^8$-$C_8H_8$ unit is, albeit the lower ring diameter, significantly closer to the neodymium ion than the $\eta^9$-$C_9H_9$ unit. This is probably caused by the higher negative charge of $\eta^8$-$C_8H_8$ compared to $\eta^9$-$C_9H_9$, leading to a stronger electrostatic attraction.

**Raman-spectroscopic analysis.** We further investigated the bonding situation and ligand aromaticity using Raman spectroscopy and vibrational analysis. Fourier transform Raman (FT-Raman) spectra of **3a–3d** were recorded powdered samples (Fig. 6). The band assignments were validated by quantum chemical calculations. The Raman spectra can be divided in two sections: (i) above 300 cm$^{-1}$ the spectra of all molecules are almost identical as the signals are solely attributable to both sandwich ligands. Vibrational coupling to the lanthanide cations is not expected due to the orthogonality of the in-plane vibrations of the ligands with respect to that of the lanthanide-ring centroid[46]. Therefore, the signals at 3042 ($\eta^8$-$C_8H_8$) and 3006 cm$^{-1}$ ($\eta^9$-$C_9H_9$) are attributed to the fully symmetric C–H-valence

motions, those at 1495 ($\eta^8$-$C_8H_8$) and 1517 cm$^{-1}$ ($\eta^9$-$C_9H_9$) to the antisymmetric C–C-stretching vibrations of the ligands. A third group of bands belongs to the (local) symmetric ring breathing modes at 749 cm$^{-1}$ ($\nu_{sym}$($\eta^8$-$C_8H_8$)) and at 681 cm$^{-1}$ ($\nu_{sym}$($\eta^9$-$C_9H_9$)). Analyzing these modes is an unambiguous proof of the ring size and the bond strength within these aromatic ligand systems. According to normal coordinate analyses on $C_5H_5^-$, $C_6H_6$ and $C_7H_7^+$ [47] the stretching force constant values of the C–H and C–C bonds are each of comparable size. Assuming, this is furthermore true for larger aromatic $C_nH_n$ monocyclic ligand systems, the approximate wavenumber of the fully symmetric ring breathing mode is easily calculated using a mathematical expression deduced in the Supplemental Information (see Supplementary Equation 1). This formula nicely reproduces the observed ring breathing mode energies of the two ligands ($\nu$($C_8H_8^{2-}$) = 754 cm$^{-1}$ (calc. 761 cm$^{-1}$) and $\nu$($C_9H_9^-$) = 680 cm$^{-1}$ (calc. 680 cm$^{-1}$)) and therefore, confirms the comparable bonding situation in these ligand systems with those of aromatic ligands like $C_5H_5^-$, $C_6H_6$ and $C_7H_7^+$ (see SI for details). (ii) At vibrational energies lower than 300 cm$^{-1}$ lanthanide-ring centroid stretching modes are expected in the Raman spectra of [($\eta^9$-$C_9H_9$)Ln($\eta^8$-$C_8H_8$)]. Due to the larger ($\eta^9$-$C_9H_9$)-Ln bond lengths their vibrations are found between 100 and 166 cm$^{-1}$, whereas those of the shorter Ln-($\eta^8$-$C_8H_8$) bonds are observed between 236 and 247 cm$^{-1}$ (Nd, **3a**: 137.0 (s), 242.1 (s); Sm, **3b**: 126.8 (s), 236.0 (vs); Dy, **3c**: 100 (s), 237.2 (s) and Er, **3d**: 165.9 (m), 207.4 (m)). Both vibrations remind of the symmetric and antisymmetric modes of triatomic linear molecules like $CO_2$. For these bands, the agreement with the results of the quantum chemical calculations is only of limited accordance due to possible coupling with lattice vibrations (see Table 4b in the SI). However, **3a–d** are rare cases in modern organometallic chemistry, in which the coordination mode of the ligands can be determined by Raman spectroscopy as sole method.

**Magnetic properties of [($\eta^9$-$C_9H_9$)Er($\eta^8$-$C_8H_8$)] (3d).** Although the single-crystal X-ray structures of **3c** and **3d** show some disorder, we conclude an almost coplanar $\eta^9$-$C_9H_9$ and $\eta^8$-$C_8H_8$ arrangement of the ligands as observed in **3a** and **3b**, based on the Raman-spectroscopic analysis. This arrangement is known to exert an equatorial ligand field, which drastically stabilizes prolate shaped $m_J$ states of lanthanide ions. The prime example for this family of trivalent lanthanide ions in terms of single molecule magnetic behavior is without doubt erbium (*vide supra*), where the equatorial ligand field exerted by a $\eta^8$-$C_8H_8$ ligand, can yield SMMs with not just large energy barriers to the relaxation of the magnetization, but also leading to open hysteresis loops at temperatures as high as 10 K[26,27,36,48]. We, therefore, carried out detailed magnetic studies on compound **3d**, to test whether the asymmetric $\eta^8$-$C_8H_8$/$\eta^9$-$C_9H_9$ ligand field enhances the magnetic anisotropy of the sandwiched erbium ion. The rationale is two-fold: (*i*) as mentioned above $\eta^8$-$C_8H_8$ ligands exert a strong equatorial ligand field, resulting in erbium compounds showing slow relaxation of the magnetization; and (*ii*) the introduction of a larger cyclic ring as $\eta^9$-$C_9H_9$ could allow a closer Er-C contact, which could increase the equatorial ligand field, therefore enhancing the anisotropic characteristics of **3d**. DC magnetic susceptibility studies of **3d** were conducted in an applied field of $H_{dc}$ = 1 kOe. At room temperature the $\chi_M T$ (*T*) ($\chi_M$ is the molar magnetic susceptibility) value is 11.25 cm$^3$ K mol$^{-1}$, in agreement with the expected value for an isolated Er (III) ion (c.f. 11.48 cm$^3$ K mol$^{-1}$ for *J* = 15/2, g$_J$ = 6/5). The moment decreases smoothly upon cooling, until ca. 6 K, where it sharply decreases to 7.13 cm$^3$ K mol$^{-1}$ at 2 K (see Supplementary Fig. 11). The abrupt drop in $\chi_M T$(*T*) indicates magnetic blocking,

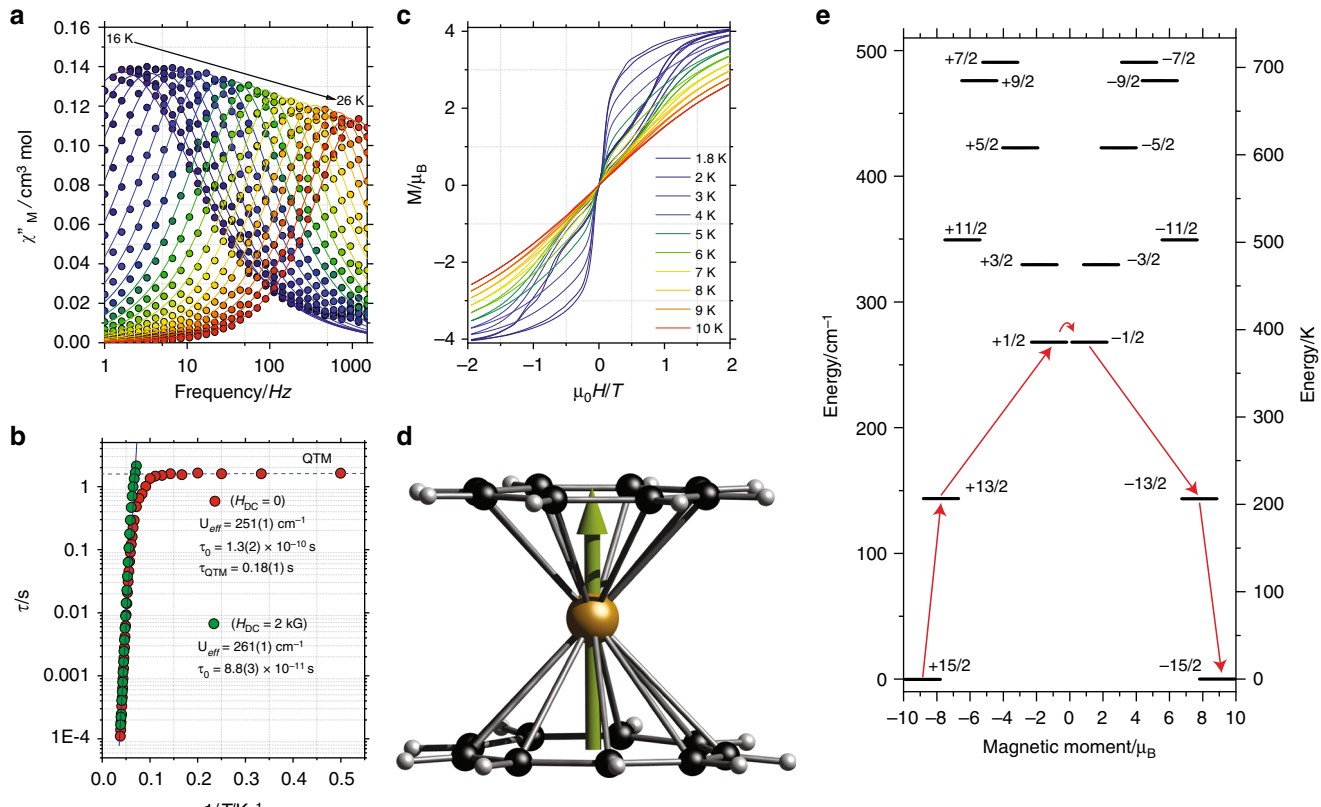

**Fig. 7** Magnetic properties of **3d**. **a** Experimental out-of-phase magnetic susceptibility ($\chi_M''(\nu)$) for **3d** in the temperature range of 16 to 26 K (filled circles). Solid lines are the fittings to a generalized Debye model. **b** Arrhenius analysis of the relaxation times ($\tau$) for **3d** with $H_{DC} = 0$ (red) and with $H_{DC} = 2$ kG (green). The black line corresponds to the thermally active regime, whilst the blue dashed line represents the QTM regime. **c** Hysteresis measurements for **3d** between ±2 T and from 1.8 K to 10 K employing a field sweep rate of 700 G/s; **d** optimized structure of **3d** and direction of the principal magnetic anisotropy axis obtained from CASSCF calculations (green arrow). **e** *ab initio* calculated electronic states of the $J = 15/2$ manifold of the $^4I$ term of **3d** and the most probable relaxation pathway for the magnetization represented by the red arrows, involving spin phonon excitation to the first and second doublets. The thick black lines represent the Kramers doublet states as function of their magnetic moments

where pinning of the magnetic moment in the immobilized crystalline material occurs.

The dynamic behavior of **3d** was probed via magnetic susceptibility AC studies under zero applied DC field. A single peak is observed in the temperature and frequency dependent out-of-phase magnetic susceptibility, i.e. $\chi_M''(T)$ and $\chi_M''(\nu)$, respectively. This result is in agreement with the dynamic studies for $[(\eta^8\text{-}C_8H_8)_2Er]^-$ [28], while they differ from the asymmetric $[(\eta^5\text{-}C_5H_5)Er(\eta^8\text{-}C_8H_8)]$ counterparts[24], where two maxima were observed. The $\chi_M''(T,\nu)$ reveals a temperature dependent maximum at temperatures between 16–26 K, whilst below 15 K the maximum in $\chi_M''(\nu)$ remains practically constant (Fig. 7a). Between 18 and 26 K, the Arrhenius analysis of $\tau$ at different temperatures show a relaxation dominated thermally activated Orbach process, whereas below 9 K temperature independent processes dominate. The curvature between 10 and 15 K suggests that other relaxations pathways, such as Raman, are also active. The distribution of the relaxation parameter ($\alpha$) likewise indicates a narrow distribution of relaxation times between 20 K and 26 K ($\alpha \leq 0.18(1)$), while for temperatures below 15 K the parameter is greater ($\alpha \geq 0.18(1)$). The energy barrier $U_{eff}$ of 251(1) cm⁻¹ and $\tau_0 = 1.3(2) \times 10^{-10}$ s (Fig. 7b) are very similar to the ones observed for homoleptic and heteroleptic erbium complexes[24,27,28,36]. The plateau at temperatures between 2 and 5 K marks the quantum tunneling of the magnetization regime, with a $\tau_{QTM} = 0.18(1)$ s. Application of an optimal field of 2 kG (at which relaxation is slower), efficiently suppressed QTM,

leading to an almost purely temperature dependent relaxation (green symbols in Fig. 7b and SI (see Supplementary Fig. 14)), with a slightly enhanced $U_{eff} = 261(1)$ cm⁻¹.

An open magnetic hysteresis is the ultimate proof of the strong anisotropic behavior in SMMs and their bistable magnetic behavior. Extrapolation of the Arrhenius data to low temperature indicates that the observation of magnetic hysteresis below 12 K is feasible, where the relaxation time is 100 s. To confirm the SMM behavior and the slow relaxation observed through AC studies, magnetization hysteresis loops were collected between 1.8–10 K. Figure 7c shows butterfly-like hysteresis loops between 2 and 10 K and a field ranging from ±2 T, leading to a blocking temperature ($T_B$) of 10 K. Note that, albeit the energy barrier being rather large, the hysteresis loops are practically close at zero field, strongly indicating that QTM is rather efficient, as commonly observed in lanthanide-based SMMs.

To gain deeper insight into the relaxation mechanism and the anisotropic magnetic properties of **3d**, CASSCF/SO-RASSI/ SINGLE_ANISO calculations were performed[49–54]. Due to the highly disordered character of the crystal structure of **3d**, CASSCF calculation were carried out employing an optimized crystal structure (see SI for details). The electronic calculation predicts a highly axial ground state with $g_z = 17.95$ and $g_{x,y} \approx 10^{-5}$. As observed in Fig. 7d, the anisotropy axis is perpendicular to the $\eta^8\text{-}C_8H_8$ and $\eta^9\text{-}C_9H_9$ planes. Employing the ligand field parameters from electronic calculations, we find that the ground, first and second excited states are almost colinear and highly pure

$m_J = \pm 15/2$ and $\pm 13/2$ and $\pm 1/2$ states, respectively. The relative energies for the first and second excited state are 140 and 268 cm$^{-1}$. The succeeding excited states are highly pure and huddled over 330–490 cm$^{-1}$. As it can be observed in Fig. 7e, *ab*-initio results reproduce rather well the $\chi_M T(T)$ and $M(H)$ (see Supplementary Fig. 11) with only small differences. The discrepancies might arise by structural distortions not reflected in the geometry optimization. Using the average matrix elements of magnetic moment between the electronic states, it is predicted that the most efficient magnetic relaxation pathway is to occur via thermally assisted QTM through the second excited state at 268 cm$^{-1}$. As observed, this state is very close to the U$_{eff}$ obtained from dynamic studies (*cf.* ~260 cm$^{-1}$ (Fig. 7e)). Interestingly, the energy barrier is also very similar to the antisymmetric vibration of the C$_8$/C$_9$ rings observed in the Raman spectrum (240 cm$^{-1}$). As molecular vibrations have been predicted to facilitate spin-phonon coupling, these could be relevant for the relaxation in **3d**[33].

Note that the strongly equatorial ligand field exerted by the $\eta^8$-C$_8$H$_8$ and $\eta^9$-C$_9$H$_9$ ligands is optimal at stabilizing the largest $m_J$ state for Er(III), characterized by a prolate electron density, as demonstrated by the magnetic studies and supported by CASSCF calculations. In contrast, for the Dy(III) ions an axial ligand field is more suitable to stabilize the largest $m_J = 15/2$ state, thus the anisotropic magnetic properties in $[(\eta^9\text{-}C_9H_9)Dy(\eta^8\text{-}C_8H_8)]$ are expectedly worse, as confirmed by AC tests and other reports[27,55].

## Discussion

By synthesizing $[(\eta^9\text{-}C_9H_9)Ln(\eta^8\text{-}C_8H_8)]$ (Ln=Nd, Sm, Dy, Er), we unveiled a fundamental class of pure sandwich complexes. The title compounds represent a long sought asymmetric organometallic motif, leading to the observation of hysteresis loops up to 10 K. In addition, we observe fast quantum tunneling of the magnetization near zero field, which opens the possibility of nuclear spin read-out with the $^{167}$Er(III) analog of $[(\eta^9\text{-}C_9H_9)Er(\eta^8\text{-}C_8H_8)]$[56]. Our results clearly highlight the significance of not just a long desired and extremely elusive organometallic complex class, but are also of relevance to future quantum spintronic applications.

## Methods

**Synthetic methods.** Experiments were carried out under a dry, oxygen-free argon atmosphere using Schlenk-line and glovebox techniques. All solvents and reagents were rigorously dried and deoxygenated before use. All compounds were characterized by single-crystal X-ray diffraction studies. Further details are available in the Supplementary Information (see section Supplementary Methods).

## Data availability

All data is available from the authors on reasonable request. The X-ray crystallographic coordinates for structures reported in this study have been deposited at the Cambridge Crystallographic Data Centre (CCDC), under deposition numbers 1894445-1894450. These data can be obtained free of charge from The Cambridge Crystallographic Data Centre via www.ccdc.cam.ac.uk/data_request/cif.

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

## Acknowledgements

K.I.T. is acknowledged for financial support. The authors acknowledge computational support by the state of Baden-Württemberg through bwHPC and the Deutsche Forschungsgemeinschaft (DFG) through grant No INST 40/467-1 FUGG.

## Author contributions

L.M. synthesized and analyzed all compounds with support from C.S. and S.B. L.M. and C.S. conducted X-ray experiments. E.M.P. and M.R. conducted and interpreted magnetic measurements and carried out the ab-initio CASSCF calculations and interpreted the results. R.K. performed and analyzed quantum chemical calculations. PWR originated the idea, supervised the work, and interpreted the results. All authors contributed to the preparation of the manuscript.

## Additional information

**Competing interests:** The authors declare no competing interests.

