## [Transparent Peer Review File · Nature Communications]

Reviewers' comments:

Reviewer #1 (Remarks to the Author):

The crystallographic part of this paper has been carefully and appropriately carried out.

All of the refined structure have been modeled in a satisfactory way.

My only doubt is regarding the Er compound: the authors report it to be a perfect (η^9 -C₉H₉)Ln(η^8 -C₈H₈) sandwich complex like the others but I am not entirely convinced that it is the case.

The disorder does not help in clarifying the issue but it seems to me that each Er atom has one ring in full η^9 or η^8 coordination whilst the other, is too "away on one side" to maintain the same coordination mode with the same Er atom.

Expanding the model to P1 removes all the symmetry and allows to visualize all of the rings at the same time and they seem to corroborate my idea.

the raman spectra could help but - as mentioned by the authors themselves - are not of great help in dealing with ligand to metal couplings.

In short, I recommend this paper to be published but I would like to have my doubt clarified first.

Reviewer #2 (Remarks to the Author):

The authors report in the synthesis of the new type of heteroleptic sandwich complexes of lanthanides with 8- and 9-membered carbon rings. The complexes are synthesized for a series of lanthanides and their molecular structures are established by single-crystal X-ray diffraction. Beautiful and illustrative Raman study of the complexes is presented. Finally, the complex with Er is shown to be a single molecule magnet and characterized by static and dynamic magnetometry. Overall, I believe these results will be of interest for inorganic and organometallic chemist as well as for scientists working on molecule magnets. I therefore recommend it for publication in Nat. Commun. after the following issues are addressed by the authors.

1. I like the Raman results very much, but I think the authors need to present their Raman study in a more clear and concise way. First, it would be important to show in the main manuscript comparison of the spectra of all 4 compounds in one figure, maybe using a limited frequency range (say, such as shown in the inset in Fig. 4). As metal atoms are rather heavy, the vibrations with their displacements should be well separated from those of the carbon rings, unless the latter exhibit translation or librations. One might be able to see a clear correlation between the mass of the lanthanide and a vibrational frequency (as an example of a similar situation, see Nat. Commun. 2019, 10, 571). Analysis in this direction could be given and will not take much effort from the authors taking into account that they have DFT calculations of vibrational spectra in hand. Besides, I recommend to add normal mode analysis for at least one of the molecule and add tables with comparison of computed and experimental frequencies in supporting information. Again, the authors have all data in hand needed for this, it is only a question of presentation of the results.

2. Discussion of magnetic properties will benefit from a comparison of Er to Dy. More importantly, the authors mention the α parameter of ca 0.4 or even more. This is a rather large value and should be at least commented. Adding tables with relaxation times to supporting information would be also beneficial for presentation of the data. Please also give the value of 100-s blocking temperature, which is not clearly defined in the manuscript. I understand that in zero field the

relaxation time is small due to the QTM, but in the finite field it can be much longer. So the measurement of relaxation times in a certain finite field would be recommended.

3. Some more technical remarks:

Fig. 3, caption: replace "on part" with "one part"

Fig. 5, caption: replace "most probably" with "most probable"

Fig. 5b: the Ueff barrier in the figure and on the page 11 is 316(1) K, but on the page 13 the value for the same barrier is 361(1) K. Please clarify which value is correct. It also affects comparison to the calculations, which predict the barrier of 386 K. If experimental value is 316 K, then the difference is rather large, but if it is 361, then the agreement is really very good.

Alexey A. Popov
IFW Dresden

Reviewer #3 (Remarks to the Author):

This paper describes the synthesis of the first structurally characterised series of unsymmetrical sandwich complexes that combine C₈H₈²⁻ and C₉H₉ anions. Four lanthanide sandwich complexes for this family have been prepared and characterized by single crystal XRD, Raman spectroscopy and magnetic measurements. Although these complexes were first targeted in the 1970s, they could not be isolated. Lanthanide metallocene-derived complexes are currently of interest as candidates for high-temperature magnetic hysteresis in single-molecule magnetism.

Although the new complexes are very nice indeed, and the paper is well written, I do not consider this work suitable for publication in Nature Comm. My reasoning is that from a synthetic point of view the complexes can be viewed as a combination of [Er(C₈H₈)₂]⁻ (ref#30), [M(C₈H₈)(C₅H₅)] (e.g. ref#21) and [Ln(C₉H₉)₂] (ref#14). Ref#14 was published in JACS 6 months ago. From a magnetic point of view, [Er(C₉H₉)(C₈H₈)] exhibits almost identical magnetic properties to [Er(C₈H₈)₂]⁻, which is essentially what would be expected based on work and theories already in the literature. By viewing this work as an important continuation of this research avenue, I would suggest a more appropriate journal may be Chem. Sci., Chem. Commun. etc.

I have noted some points for the authors to consider:

1. Abstract: I suggest that a better word than linear would be axial. Linear suggests the presence of a line and this is only defined by the centroids, so this word can only be used if this is made explicit.

2. Introduction: I am unclear why the authors chose chromocene as an example of a well-established sandwich complex – I think that perhaps they mean to cite bis(benzene)chromium, which at least has a different motif to ferrocene and the MCp₂ metallocene family. Also, by reducing the latter part of the literature search to ring numbers of x = 5-9 the authors have missed out [Co(C₅H₅)(C₄H₄)] (J. Organomet. Chem. 1976, 113, 157-166); as this is only one extra example for x = 3-9 I think they should redefine the range of the search add this to the references and to Figure 1 (indeed a different substituted Co C₄/C₅ combination has been cited). Otherwise this section is well written and has provided good context.

3. Results and discussion:

- The authors indicate that the C₈ ring being closer than the C₉ ring in the solid state structure of 3a is unusual given the smaller ring diameter, but given the additional electrostatic component of the -2 C₈ ring vs. the -1 C₉ ring this is not surprising, and this sentence should be modified to incorporate this point.
- Minor point, three atomic > triatomic.
- Prolate shaped lanthanide ions are referred to: the ions do not have prolate shapes, they are

spherical. More precisely some ions have prolate-spheroid shaped magnetic mJ states, and stabilization of these states leads to favourable SMM properties; please change to clarify. A citation to the 10 K Er SMM (#31) should be added to this paragraph.

- From page 11 onwards there appears to be a copy and paste error, where the authors refer to the magnetic properties of 3b but this is the Sm complex. This must still be referring to 3d, which is the Er complex, as the authors discuss the mJ 15/2 state.

4. Conclusions: the authors state that the MC8/C9 family may be the last isolatable member of pure sandwich complexes. I think that this comment should be removed, as I see no reason why e.g. C7/C9, C5/C9 should not be isolatable based on previous work. I also think that the last comment on the demonstration of the importance of this work in future spintronic applications should be softened to "is also of relevance to" as the link is not direct.

5. Supporting Information: Please include the elemental analysis data even if the carbon values are low, to show your data. I also think that EPR data (for Nd and Er) and electronic spectra for all complexes would greatly improve the thoroughness of this study.

Reviewer #4 (Remarks to the Author):

This is a very interesting study on lanthanide sandwich complexes consisting of a cyclononatetraenyl ligand and a COT ligand. The new compounds represent something of a milestone in organometallic chemistry and will be well-received by the wider community. The erbium version was studied in the context of single-molecule magnet (SMM) properties, and found to fit a popular theoretical model, which provides important experimental support for the relevant arguments. The Raman spectroscopy is a nice addition.

Publication is recommended subject to a series of mostly minor revisions.

This Reviewer's interests in organometallic sandwich compounds may be a disadvantage in assessing the intro because, to me, it appears almost too general, particularly the comments on ferrocene, chromocene, uranocene, etc., which are of borderline relevance. With the current approach, there is also no obvious reason to omit s- and p-block metallocenes.

Given the remit of the journal, I appreciate the need to appeal to a wider audience, however I feel that the new results could be framed more appropriately if the recent findings with organometallic sandwich SMMs were summarized in a more focused way – this is the aspect that the general reader may not fully appreciate. For example, adding a short discussion of the respective oblate and prolate natures of the 4f electron density of dysprosium and erbium and, hence, the reason why axial and equatorial crystal fields, respectively, are appropriate for targeting SMM properties. The authors already cite the relevant papers, so this shouldn't require much additional work.

At the bottom of p3 of the manuscript, there is a mention of two Co complexes and a reference to Fig. 1. These complexes do not appear in the Figure. It was also not immediately clear to me what the (Ph-BPh3) ligand might be.

Figure 1 caption. What is M? Could it be a metal from the s-, p-, d- or f-block?

It would help to define the term 'sandwich complex'. Just two pi-bonded ligands? Does the definition include Cp4M compounds?

SMM = single-molecule magnet, not single-molecular magnet.

The comment on p4 about the role of local symmetry is valid but some of the associated citations do not seem appropriate. Ref 41 is fine, but refs 39, 40 and 42 are curious. I would recommend replacing them with: Ungur L, Chibotaru LF (2016) *Inorg Chem* 55:10043; Liu JL, Chen YC, Tong, ML (2018) *Chem Soc Rev* 47:2431; Rinehart JD, Long JR (2011) *Chem Sci* 2:2078.

P5: delete 'CH' on line 4.

The description of the structure of the C9 ring as being 'circular' should be changed to something that describes a regular nonagon. The statement 'Only the circular all cis-isomer was found in the solid state and no positional disorder indicating the presence of the cis,cis,cis,trans-cyclononatetraenyl isomer was observed' also needs to be resolved because it currently means that an isomer with a trans component was actually found.

I wonder why the authors went to such lengths to synthesize [(COT)LnI(thf)_n] when they can be obtained using a recent protocol adapted by Rinehart et al, as cited in Ref 46 and other papers from the same group.

Since we are seeing a homologous series of Ln-C9 complexes, it would be good to mention the metal-centroid distances in the main manuscript.

The text on p8 mentions a bonding analysis involving DFT. I couldn't locate this in the ms or the SI.

A table of the Raman data would be very helpful.

Given the current popular model of spin-phonon coupling and its role in facilitating Orbach relaxation, was it possible to identify none-stretching modes in the Raman spectra, e.g. out-of-plane deformations of the C9/C8 rings? Of course, it could be the case that these modes are Raman-inactive.

Regarding the statement 'the composition can be determined by Raman spectroscopy as sole method', is this intended to mean that, given the spectrum without prior knowledge of the identity of the sample, it should be possible to determine the formula of the compound? Please clarify the meaning.

The comment about the prolate nature of erbium on p10 requires some citations (Murugesu, Rinehart, Long, Gao), as does the sentence at the end of the paragraph in question.

The statement 'resulting in slow relaxing erbium-based SMMs' should be modified to something along the lines of 'resulting in erbium compounds showing slow relaxation of the magnetization' or 'erbium-based SMMs'.

By 'larger equatorial ligand field' do the authors mean a stronger field? How is the proposed stronger nature of the C9 crystal field reconciled with the lower electrostatic charge of this ligand relative to [COT]²⁻?

Fig S9: can the sharp drop at 6 K also be an indication of the onset of magnetic blocking? There is also a small increase in XT at very low T, which is not reproduced by the CASSCF calcs: is this just a quirk of the SQUID magnetometer? Such artefacts are well-known for Quantum Design instruments so it isn't a problem, but it should be mentioned in the caption.

If I look at the X''(T) data in Fig S10, I think I can see the emergence of a second peak in the data at T approx. = 12-14 K. Can the authors comment on this? The same feature doesn't appear in the X''(ν) data so this could again be the SQUID artefact (or the reviewer's eyesight). However, the authors do mention that other relaxation pathways could be active at T = 10-15 K. The large

alpha-parameter may help here.

Personal preference: U_{eff} is better in units of cm^{-1} . Do the authors think the pre-exponential factor is reasonable?

Fig. 5 caption: the most probable relaxation pathway is mentioned twice (one with 'probably'). Replace 'phono' with 'phonon'.

Conclusions: I would counsel caution in relation to the statement about the compounds in this study being the last type of isolable sandwich compound, which is bold and likely to be disproved.

The authors could consider a short comment on the fact that current study also provides further support for the prolate model of the electron density in erbium SMMs, which is a nice result.

Reviewer #1:

1. My only doubt is regarding the Er compound: the authors report it to be a perfect ($\eta^9\text{-C}_9\text{H}_9$)Ln($\eta^8\text{-C}_8\text{H}_8$) sandwich complex like the others but I am not entirely convinced that it is the case. The disorder does not help in clarifying the issue but it seems to me that each Er atom has one ring in full η^9 or η^8 coordination whilst the other, is too "away on one side" to maintain the same coordination mode with the same Er atom. Expanding the model to P1 removes all the symmetry and allows to visualize all of the rings at the same time and they seem to corroborate my idea.

Answer: We appreciate the referee's comment on the Er compound's structure. Indeed, expanding the model to P1 does allow a better visualization of the molecular solid-state structure of **3d** (a CIF file of our solution and refinement in P1 is attached to this letter). Moreover, the referee is correct, that the molecular crystal structure of this compound does not doubtlessly proof its perfect sandwich type configuration. We therefore added a few sentences on the solid-state structure of **3d** to the MS, that should address this problem. Yet, we do think, that the further analytics we performed hint towards **3d** being a sandwich type compound, as proposed initially. First and foremost, geometry optimization results in such a configuration. Secondly the Raman spectra indicate that both ($\eta^9\text{-C}_9\text{H}_9$)⁻ and ($\eta^8\text{-C}_8\text{H}_8$)²⁻ moieties of **3d** are in a similar environment, as observed in compounds **3a-c**, whereas the sandwich configuration of these compounds could unquestionably be proofed via XRD. However, the Raman spectra are no final proof, as the referee already mentioned in the comment. Therefore, CAASCF calculations were performed with the optimized structure of **3d** and nicely predict its magnetic properties, which can be considered an evidence for the significance of the optimized, sandwich type structure.

Reviewer #2:

1. I like the Raman results very much, but I think the authors need to present their Raman study in a more clear and concise way. First, it would be important to show in the main manuscript comparison of the spectra of all 4 compounds in one figure, maybe using a limited frequency range (say, such as shown in the inset in Fig. 4).

Answer: A further figure comprising of experimental and theoretical Raman spectra of compounds **3a-d** was included in the SI (Figure S10); the frequency range covers the metal ring stretching modes and the ring tilt motions of **3a** to **3d** from 780 to 60 cm^{-1} . We think that this Figure would exceed the scope of the manuscript and therefore put it into the SI. Of course, this can still be changed.

2. As metal atoms are rather heavy, the vibrations with their displacements should be well separated from those of the carbon rings, unless the latter exhibit translation or librations. One might be able to see a clear correlation between the mass of the lanthanide and a vibrational frequency (as an example of a similar situation, see Nat. Commun. 2019, 10, 571).

Answer: The vibrations of the ring systems are properly separated from those of the metal carbon system, but a correlation of the latter signals with the masses of the metal atoms cannot be confirmed. The data are summarized in the following table:

Table 1: Metal-Ring vibrations of approximately symmetric and antisymmetric character of **3a-d**.

	„sym.“ metal-ring vibration	„antisym.“ metal-ring vibration
	$\nu_{\text{exp.}} (\nu_{\text{theor.}})/\text{cm}^{-1}$	$\nu_{\text{exp.}} (\nu_{\text{theor.}})/\text{cm}^{-1}$
Nd (3a)	137.0 (131.1)	242.1 (221.7)
Sm (3b)	126.8 (128.5)	236.0 (197.7)
Dy (3c)	100 (127.7)	237.2 (222.4)
Er (3d)	165.9 (130.9)	207.4 (190.4)

We do not expect such a behavior due to the following reasons:

(i) According to the low symmetry (C_s) of the molecules many couplings are expected and thus an "unambiguous" assignment of a band to a (metal-carbon ring) motion is often not possible.

(ii) Furthermore, in the low frequency range, different and poorly estimable couplings to lattice vibrations occur. This is especially evident in the quasi-"symmetric" metal-ring modes, in which the metal atoms move very little and thus should have only a negligible influence on the vibration frequency (see Table 1). This expectation is confirmed for the theoretical (gas phase) values, but not for those deduced from the measurement of the solids.

(iii) The experimental values of the quasi-"antisymmetric" metal-ring modes show a greater influence of the mass of the metal atoms, but the behavior expected by the reviewer is as well not confirmed by the theoretical results – probably due to different couplings caused by the low symmetry.

3. Analysis in this direction could be given and will not take much effort from the authors taking into account that they have DFT calculations of vibrational spectra in hand.

Answer: This analysis was performed on the Sm compound **3b** by means of vibrational visualization obtained from the Eigen vector data of the DFT frequency calculation. Both a table (Table S4) and a figure (Figure S9) were included into the SI.

4. Besides, I recommend to add normal mode analysis for at least one of the molecule and add tables with comparison of computed and experimental frequencies in supporting information. Again, the authors have all data in hand needed for this, it is only a question of presentation of the results.

Answer: We fully agree with the referee that a normal coordinate analysis - also because of the already existing theoretical force constant matrix - represents the perfect way to determine the forces in this type of molecule. Valence force constants are an excellent measure to evaluate the bond strength: they make a direct statement of how much force is necessary to tie two bonding partners, so they vibrate! Although the target molecules discussed here look beautifully symmetrical, they unfortunately possess only C_s symmetry, so that there are already 51 internal coordinates in the fully symmetric irreducible representation with e.g. 9 different internal metal-carbon valence force constants. We were unsuccessful to perform such a calculation due to its size.

5. Discussion of magnetic properties will benefit from a comparison of Er to Dy. More importantly, the authors mention the α parameter of ca 0.4 or even more. This is a rather large value and should be at least commented. Adding tables with relaxation times to supporting information

would be also beneficial for presentation of the data. Please also give the value of 100-s blocking temperature, which is not clearly defined in the manuscript. I understand that in zero field the relaxation time is small due to the QTM, but in the finite field it can be much longer. So the measurement of relaxation times in a certain finite field would be recommended.

Answer: We added a sentence to the manuscript, shortly pointing out the “worse” magnetic properties of the Dy compound **3c**. This is in agreement with the popular model of stabilizing m_J states with the aid of an appropriately shaped ligand field. The referee is correct, and the alpha parameter is large, indicating that further relaxation pathways are operative contributing to the relaxation dynamics of the SMM. In addition, the temperature independent regime (QTM) is reached at temperatures below 15 K. As the referee points out, QTM can be reduced by application of an additional static DC field, which commonly brings the system away from crossings, where QTM is operative. We have followed the referees advise and after determination of an optimal field of 2 kG (field at which the relaxation is slower), we have conducted the relaxation studies at diverse frequencies and temperatures. Firstly, we find that QTM is successfully suppressed, as observed in the $\tau(T)$ (cf. to the $\tau(T)$ at zero field). This is reflected in the almost linear $\tau(T)$ with a small curvature just below 16 K, leading to $U_{eff} = 376(2)$ K. Note that although application of a DC field quenches the QTM, other relaxation pathways (such as Raman processes) might still be operative, as observed in the curvature of $\tau(T)$ below 16 K. We have added the relaxation times, alpha parameters, isothermal and adiabatic magnetic susceptibilities, as well as images of the new data sets to the SI and reflected all changes in the MS. Additionally, the 100-s blocking temperature has been clarified in the MS.

6. Some more technical remarks: Fig. 3, caption: replace “on part” with “one part”

Answer: The typo was corrected.

7. Fig. 5, caption: replace “most probably” with “most probable”

Answer: The word was corrected.

8. Fig. 5b: the U_{eff} barrier in the figure and on the page 11 is 316(1) K, but on the page 13 the value for the same barrier is 361(1) K. Please clarify which value is correct. It also affects comparison to the calculations, which predict the barrier of 386 K. If experimental value is 316 K, then the difference is rather large, but if it is 361, then the agreement is really very good.

Answer: We apologize to the referees, indeed there is a mistake on the U_{eff} . We have re-checked the data and the correct value is 362(1) K and has been corrected in the whole MS. Note, that we changed all values from K to cm^{-1} . Therefore, the value now found in SI and manuscript is 251(1) cm^{-1} .

Reviewer #3

1. Abstract: I suggest that a better word than linear would be axial. Linear suggests the presence of a line and this is only defined by the centroids, so this word can only be used if this is made explicit.

Answer: We substituted linear by sandwiched.

2. Introduction: I am unclear why the authors chose chromocene as an example of a well-established sandwich complex – I think that perhaps they mean to cite bis(benzene)chromium, which at least has a different motif to ferrocene and the MCp2 metallocene family. Also, by reducing the latter part of the literature search to ring numbers of $x = 5-9$ the authors have missed out [Co(C₅H₅)(C₄H₄)] (J. Organomet. Chem. 1976, 113, 157-166); as this is only one extra example for $x = 3-9$ I think they should redefine the range of the search add this to the references and to Figure 1 (indeed a different substituted Co C₄/C₅ combination has been cited). Otherwise this section is well written and has provided good context.

Answer: The reviewer is right in both cases. Bis(benzene)chromium is a way better example, and we missed [Co(C₅H₅)(C₄H₄)]. Thus Figure 1 was revised.

3. Results and discussion:

The authors indicate that the C₈ ring being closer than the C₉ ring in the solid state structure of 3a is unusual given the smaller ring diameter, but given the additional electrostatic component of the -2 C₈ ring vs. the -1 C₉ ring this is not surprising, and this sentence should be modified to incorporate this point.

Answer: The reviewer's comment is correct. We therefore adapted the sentence to better reflect this point.

4. Minor point, three atomic > triatomic.

Answer: We substituted three atomic with triatomic

5. Prolate shaped lanthanide ions are referred to: the ions do not have prolate shapes, they are spherical. More precisely some ions have prolate-spheroid shaped magnetic mJ states, and stabilization of these states leads to favourable SMM properties; please change to clarify. A citation to the 10 K Er SMM (#31) should be added to this paragraph.

Answer: We corrected this inaccuracy in the manuscript.

6. From page 11 onwards there appears to be a copy and paste error, where the authors refer to the magnetic properties of 3b but this is the Sm complex. This must still be referring to 3d, which is the Er complex, as the authors discuss the mJ 15/2 state.

Answer: Of course, the referee is right here. It must be compound **3d**, as this is the compound being discussed. We corrected this error.

7. Conclusions: the authors state that the MC₈/C₉ family may be the last isolatable member of pure sandwich complexes. I think that this comment should be removed, as I see no reason why e.g. C₇/C₉, C₅/C₉ should not be isolatable based on previous work. I also think that the last comment on the demonstration of the importance of this work in future spintronic applications should be softened to "is also of relevance to" as the link is not direct.

Answer: We removed the statement about the "last isolatable member" and edited the last sentence.

8. Supporting Information: Please include the elemental analysis data even if the carbon values are low, to show your data. I also think that EPR data (for Nd and Er) and electronic spectra for all complexes would greatly improve the thoroughness of this study.

Answer: We thank the referee for the comment. We therefore performed exemplary UV/VIS measurements for compounds **3b** and **3d** (see Figure 1). Except for the band at approximately 285 cm^{-1} no resonance is observable in both cases. We assume, that interesting absorption bands of these compounds are only present below 280 nm. An area not accessible due to the UV cutoff of toluene and the quartz cuvettes used for the measurement.

Figure 1: UV/VIS spectra of compounds **3b** (left; $c = 5.38 \cdot 10^{-8}$ mol/L) and **3d** (right; $c = 5.15 \cdot 10^{-8}$ mol/L) in toluene.

We think that publishing EA data of extremely air sensitive compounds, that moreover are badly combustible, is not very sensible. On the same note, we also do not show NMR data of these compounds due to their insolubility. Concerning the EPR comments, EPR spectroscopy concerns with electronic transitions between electronic states with a $\Delta m_s = \pm 1$. For common EPR spectrometers just the ground doublet is accessible (X-band = ~ 9 GHz (0.3 cm^{-1}), Q-band ~ 34 GHz (1 cm^{-1}) and W-band ~ 90 GHz (3 cm^{-1})). In our work, we show that, for the most interesting compound, i.e. $[(\eta^9\text{-C}_9\text{H}_9)\text{Er}(\eta^8\text{-C}_8\text{H}_8)]$, the m_J ground state is $m_J = 15/2$ with a large separation between the ground and first excited state (more than 200 K, i.e. 144 cm^{-1}). Now, for the ground doublet of **3d**, the $\Delta m_J = 15$, is expected to render this complex EPR silent in common spectrometers. High-Field-High-Frequency EPR might provide the required operating frequencies; however, such studies would be the focus of more dedicated future work. Similar arguments, as for Er, are valid for the Nd complex, in which preliminary theoretical studies also predict a large m_J state as ground state and large separation from the first excited state, while for the Sm(III) analogue, an m_J with the lowest multiplicity is expected. Indeed, the referee is correct and such studies might be of interest, however we consider such work for a more dedicated study in the near future, given that main objective in our current manuscript is the report of synthesis a new family of long-sought complexes.

Reviewer #4

1. This Reviewer's interests in organometallic sandwich compounds may be a disadvantage in assessing the intro because, to me, it appears almost too general, particularly the comments on ferrocene, chromocene, uranocene, etc., which are of borderline relevance. With the current approach, there is also no obvious reason to omit s- and p-block metallocenes. Given the remit

of the journal, I appreciate the need to appeal to a wider audience, however I feel that the new results could be framed more appropriately if the recent findings with organometallic sandwich SMMs were summarized in a more focused way – this is the aspect that the general reader may not fully appreciate. For example, adding a short discussion of the respective oblate and prolate natures of the 4f electron density of dysprosium and erbium and, hence, the reason why axial and equatorial crystal fields, respectively, are appropriate for targeting SMM properties. The authors already cite the relevant papers, so this shouldn't require much additional work.

Answer: We appreciate the referee's comment. We intended to stress the possible importance of the reported compounds in the context of the fundamental class of pure metalorganic sandwich complexes, thus, we initially didn't focus much in the magnetic properties in the MS. We have now added few sentences dealing with oblate and prolate nature of the 4f electron density to the MS.

2. At the bottom of p3 of the manuscript, there is a mention of two Co complexes and a reference to Fig. 1. These complexes do not appear in the Figure. It was also not immediately clear to me what the (Ph-BPh₃) ligand might be.

Answer: The referee is right here. There was a lack of clarity considering the examples for sandwich complexes in the introduction. The two Co complexes are not depicted in Fig.1, as Fig.1 only shows reported structural motives comprising only unsubstituted C_xH_x-rings. We therefore adapted the choice of examples and edited Fig.1 accordingly.

3. Figure 1 caption. What is M? Could it be a metal from the s-, p-, d- or f-block?

Answer: We added in the caption (M = metal atom).

4. It would help to define the term 'sandwich complex'. Just two pi-bonded ligands? Does the definition include Cp₄M compounds?

Answer: We defined the term sandwich complexes as complexes bearing two planar, cyclic and pi-bonded ligands, to match the reviewer's remark.

5. SMM = single-molecule magnet, not single-molecular magnet.

Answer: We corrected the typo.

6. The comment on p4 about the role of local symmetry is valid but some of the associated citations do not seem appropriate. Ref 41 is fine, but refs 39, 40 and 42 are curious. I would recommend replacing them with: Ungur L, Chibotaru LF (2016) *Inorg Chem* 55:10043; Liu JL, Chen YC, Tong, ML (2018) *Chem Soc Rev* 47:2431; Rinehart JD, Long JR (2011) *Chem Sci* 2:2078.

Answer: We added the references.

7. P5: delete 'CH' on line 4.

Answer: We corrected that.

8. The description of the structure of the C₉ ring as being 'circular' should be changed to something that describes a regular nonagon. The statement 'Only the circular all cis-isomer was found in the solid state and no positional disorder indicating the presence of the cis,cis,cis,trans-

cyclononatetraenyl isomer was observed' also needs to be resolved because it currently means that an isomer with a trans component was actually found.

Answer: We changed the sentence into: Only the perfectly nonagonal all cis-isomer was found in the solid state and no positional disorder indicating the presence of the *cis,cis,cis,trans*-cyclononatetraenyl isomer was observed. Considering the second part of the reviewer's comment, we would like to point out that priorly only a mixture of the *cis,cis,cis,cis*-cyclononatetraenyl and *cis,cis,cis,trans*-cyclononatetraenyl isomers was crystallographically characterized by Nocton *et. al.* Thus we wanted to highlight the successful isolation and characterization of the prior one in its pure form.

9. I wonder why the authors went to such lengths to synthesize [(COT)LnI(thf)_n] when they can be obtained using a recent protocol adapted by Rinehart *et al.*, as cited in Ref 46 and other papers from the same group.

Answer: We wanted to emphasize, that the synthetic protocol we applied here, in contrast to the approach used by Rinehart *et. al.*, does not rely on the application of K₂COT. K₂COT tends to violently react with air, which poses a significant safety issue when prepared on a gram scale. Moreover, no expensive lanthanide triiodides are mandatory. Thus, a variety of [(COT)LnI(thf)_n] complexes can be synthesized on a rather large scale in a very convenient way.

10. Since we are seeing a homologous series of Ln-C9 complexes, it would be good to mention the metal-centroid distances in the main manuscript.

Answer: A table with the Ln-Cg distances was added to the Ms.

11. The text on p8 mentions a bonding analysis involving DFT. I couldn't locate this in the ms or the SI.

Answer: In fact, we performed a vibrational analysis. We replaced this statement in the manuscript.

12. A table of the Raman data would be very helpful.

Answer: We included additional tables into the SI, consisting of the theoretical vibrational energies and the Raman intensities (Table S2 and Table S3). Furthermore, a figure comparing the theoretical and experimental Raman spectra of the Sm compound **3b** was added to the SI (Figure S9). Moreover, a table comparing theoretical and experimental values for the same compound in detail was added (Table S4). In this table, the motion characters of the bands are included.

13. Given the current popular model of spin-phonon coupling and its role in facilitating Orbach relaxation, was it possible to identify none-stretching modes in the Raman spectra, e.g. out-of-plane deformations of the C9/C8 rings? Of course, it could be the case that these modes are Raman-inactive.

Answer: Out-of-plane deformation of the C9/C8 rings are indeed observed in the Raman spectrum with energies between 130 to 240 cm⁻¹ (symmetric and antisymmetric, respectively). Interestingly, the antisymmetric vibration is very similar to the separation of the ground state and second excited state

observed in the Erbium complex, i.e. 268 cm^{-1} . A more detailed study, focusing in the identification of the local vibrational mode and the associated crystal field parameter affected by the vibrational motion is out of the scope of our present work, however, we include a note highlighting the possibility of this vibration being responsible for the relaxation through the second excited state.

14. Regarding the statement 'the composition can be determined by Raman spectroscopy as sole method', is this intended to mean that, given the spectrum without prior knowledge of the identity of the sample, it should be possible to determine the formula of the compound? Please clarify the meaning.

Answer: Indeed, the statement was a bit provocative. We changed it in the following way: 'However, **3a-d** are rare cases in modern organometallic chemistry, in which the coordination mode of the ligands can be determined by Raman spectroscopy as sole method.' Moreover, we editet the corresponding phrase in the abstract accordingly.

15. The comment about the prolate nature of erbium on p10 requires some citations (Murugesu, Rinehart, Long, Gao), as does the sentence at the end of the paragraph in question.

Answer: We added the corresponding citations.

16. The statement 'resulting in slow relaxing erbium-based SMMs' should be modified to something along the lines of 'resulting in erbium compounds showing slow relaxation of the magnetization' or 'erbium-based SMMs'.

Answer: We changed the phrase in question.

17. By 'larger equatorial ligand field' do the authors mean a stronger field? How is the proposed stronger nature of the C9 crystal field reconciled with the lower electrostatic charge of this ligand relative to [COT]2-?

Answer: The referee is correct. We changed the sentence to better reflect the fact C_9H_9^- is bigger than $\text{C}_8\text{H}_8^{2-}$ and thus might approach a lanthanide ion closer, than the latter one, although this is not the fact, as shown by XRD-data.

18. Fig S9: can the sharp drop at 6 K also be an indication of the onset of magnetic blocking? There is also a small increase in XT at very low T, which is not reproduced by the CASSCF calcs: is this just a quirk of the SQUID magnetometer? Such artefacts are well-known for Quantum Design instruments so it isn't a problem, but it should be mentioned in the caption.

Answer: Indeed, the sharp drop at 6 K, indicates the onset of magnetic blocking, where the magnetic moment of the spins are pinned in the crystals due to the barrier to reorientation of the magnetic moment. Additionally, indeed there is a small bump at low temperature. We have re-checked this and we found this to be reproducibile for the same sample. We consider this to be an effect of weak intermolecular interactions, which are not contemplated in the CASSCF calculations. An explanation of this has been added to the SI.

19. If I look at the $\chi''(T)$ data in Fig S10, I think I can see the emergence of a second peak in the data at T approx. = 12-14 K. Can the authors comment on this? The same feature doesn't

appear in the $X''(\nu)$ data so this could again be the SQUID artefact (or the reviewer's eyesight). However, the authors do mention that other relaxation pathways could be active at $T = 10-15$ K. The large α -parameter may help here.

Answer: The referee is correct and there seems to be a broad small maximum between 12-14 K, which is solely clear in the $X''(T)$, while it is not apparent in the $X''(\nu)$. The effect of this unresolved maximum is to widen the $X''(\nu)$ traces, thus leading to a higher α parameter during the fitting procedure. We have clarified this in the SI and MS.

20. Personal preference: U_{eff} is better in units of cm^{-1} . Do the authors think the pre-exponential factor is reasonable

Answer: The U_{eff} unit has been changed to wavenumber. Also, the pre-exponential value is in the range for the observed for similar systems (see for example ref. *Angew. Chem. Int. Ed.* **53**, 4413-4417 (2014); *J. Am. Chem. Soc.* **133**, 4730-4733 (2011). *Inorg. Chem.* **2012**, *51*, 3079-3087).

21. Fig. 5 caption: the most probable relaxation pathway is mentioned twice (one with 'probably'). Replace 'phono' with 'phonon'.

Answer: We corrected the typo.

22. Conclusions: I would counsel caution in relation to the statement about the compounds in this study being the last type of isolable sandwich compound, which is bold and likely to be disproved.

Answer: We removed the sentence.

23. The authors could consider a short comment on the fact that current study also provides further support for the prolate model of the electron density in erbium SMMs, which is a nice result.

Answer: The respective comments have been added to the MS on the pages 8 and 10.

REVIEWERS' COMMENTS:

Reviewer #1 (Remarks to the Author):

I am happy with the explanation provided and the modifications in the manuscript.

as far as I am concerned the manuscript can be published

Alessandro Prescimone

Reviewer #2 (Remarks to the Author):

The authors addressed my comments and updated the manuscript. I recommend the paper for publication. The only thing which requires the attention is the list of calculated frequencies in SI, which show one "negative" frequency for each compound - this should be commented or corrected.

Alexey Popov, IFW Dresden